# An Insight into the Potassium Currents of hERG and Their Simulation

**DOI:** 10.3390/molecules28083514

**Published:** 2023-04-16

**Authors:** Rolando Guidelli

**Affiliations:** Retired Professor, Department of Chemistry “Ugo Schiff”, Florence University, Via della Lastruccia 3, Sesto Fiorentino, 50019 Firenze, Italy; rolando.guidelli@libero.it; Tel.: +39-055-457-3105

**Keywords:** hERG potassium channel, *Shaker* potassium channel, stochastic models, deterministic models, depolarization, repolarization, conformational states, cardiac action potential

## Abstract

By assuming that a stepwise outward movement of the four S4 segments of the hERG potassium channel determines a concomitant progressive increase in the flow of the permeant potassium ions, the inward and outward potassium currents can be simulated by using only one or two adjustable (i.e., free) parameters. This deterministic kinetic model differs from the stochastic models of hERG available in the literature, which usually require more than 10 free parameters. The K^+^ outward current of hERG contributes to the repolarization of the cardiac action potential. On the other hand, the K^+^ inward current increases with a positive shift in the transmembrane potential, in apparent contrast to both the electric and osmotic forces, which would concur in moving K^+^ ions outwards. This peculiar behavior can be explained by the appreciable constriction of the central pore midway along its length, with a radius < 1 Å and hydrophobic sacks surrounding it, as reported in an open conformation of the hERG potassium channel. This narrowing raises a barrier to the outward movement of K^+^ ions, inducing them to move increasingly inwards under a gradually more positive transmembrane potential.

## 1. Introduction

The human hERG K^+^ channel (Kv11.1) plays an important role in the repolarization of the cardiac action potential [1]. Many non-antiarrhythmic drugs were found to inhibit this channel determining the so-called ‘QT prolongation’, i.e., an anomalously long time spent by the heart muscle to contract and relax. Consequently, the study of in vitro interactions of drugs with hERG has become an important part of ‘safety pharmacology’. QT prolongation, in itself, is not toxic; nonetheless, this phenomenon has often been observed in humans just prior to the onset of fatal cardiac events, such as ventricular arrhythmias leading to sudden cardiac arrest and death [2]. This concomitance has been a major cause of several drug withdrawals and of the implantation of cardioverter-defibrillators in the chest to prevent this ‘sudden death’. 

Even though hERG contains a K^+^-selective pore and four voltage sensing domains like the *Shaker* K^+^ channel and other voltage-gated K^+^ channels, its gating properties and functional activity are quite different [3,4]. Both *Shaker* and hERG consist of four unconnected ‘subunits’ of high internal homology, each composed of six membrane-spanning *α*-helices (S1–S6), which are circularly arranged around a transmembrane central pore filled with water. The four S1–S4 helices of each subunit form a voltage sensing domain, with the S4 helix bearing seven positive charges in *Shaker* [5] and five in hERG [3], and with the totality of the three other helices bearing a net negative charge. The three S1–S3 helices of each voltage sensing domain are arranged to form a hydrophobic gasket called the ‘gating pore’, along which the corresponding S4 segment can move. The gating pore is interposed between two hydrophilic vestibules. Each S4 segment is covalently linked to the corresponding S5 segment by an S4–S5 linker located on the intracellular side of the membrane, as shown in Figure 1. In turn, on the extracellular side of the membrane, each S5 segment is covalently linked to the corresponding S6 segment by a loop (the ‘P loop’), which is partially folded back into the central pore, lining its upper portion and forming a funnel-like vestibule whose narrow section constitutes the ‘selectivity filter’. The latter is so called because it contains a highly conserved amino acid sequence (the ‘signature sequence’), whose carbonyl oxygen atoms can replace the hydration shell of the permeant cation, allowing its flow along the selectivity filter.

The lumen of the transmembrane ‘central pore’ is lined by the S6 segments of the four subunits, which are flanked by the corresponding S5 segments non-covalently connected to them by many amino acid contacts throughout the entire membrane. The central pore is filled with water and shows a widening at about half its full length, called the ‘central cavity’. The S6 segments are widely separated near the extracellular membrane surface, while converging at the intracellular membrane surface, where they form a sort of bundle crossing, called ‘activation gate’, which may be open or closed. Differently from *Shaker*, under the circumstances specified below, the hERG potassium channel assumes a peculiar open conformation characterized by a central pore with an atypically small cross section of radius < 1 Å. This appreciable narrowing of the central pore occurs midway along its length, as revealed by a cryo-EM structure of the open hERG [3]; it is surrounded by four deep hydrophobic pockets, which may explain the channel unusual sensitivity to many drugs. 

A sufficiently positive transmembrane potential step from a far negative ‘resting’ (‘holding’) transmembrane potential *ϕ* determines a movement of the four S4 segments from an ‘in’ to an ‘out’ position with respect to the extracellular fluid. The total number of positive charges of the four S4 segments that pass across the corresponding gating pores following their outward movement constitutes the overall ‘gating charge’. Similarly to the *Shaker* K^+^ channel [6], the outward movement of each S4 helix of hERG is transmitted through its intracellular S4–S5 linker to the corresponding S5 helix, which in turn shifts the respective S6 helix slightly away from the axis of the central pore. The activation gate is completely closed (C) when all four S4 segments are in the ‘in’ position and completely open (O) when they are all in the ‘out’ position. In moving outwards, each S4 segment rotates to form transient ion pairs with the negative charges located in the corresponding S1–S3 segments, according to the so-called ‘sliding helix model’ [7]. 

A depolarizing pulse induces the hERG K^+^ channel to move the S4 segments outwards with respect to the extracellular fluid. This movement is about two orders of magnitude slower than in the *Shaker* K^+^ channel [8] and is referred to as ‘*slow activation*’. By attaching a fluorescent probe to the extracellular S3–S4 linker of hERG, Smith and Yellen [9] verified that the kinetics of the fluorescence changes is low and correlates well with the kinetics of the slow activation of the K^+^ outward current. This provides clear evidence that the kinetics of the slow activation must be attributed to the slow outward S4 movements. This behavior is at least partially ascribable to the S1–S3 segments of hERG being more negatively charged than those of *Shaker* [10], thus exerting a stronger electrostatic attraction upon the corresponding S4 segments. Tryptophan scanning mutagenesis demonstrated that each S4 segment is loosely packed within its own voltage sensing domain [11]. The slow activation is followed by a ‘*fast inactivation*’, probably ascribed to a subtle rearrangement in the hERG selectivity filter with respect to *Shaker* [3]. This implies that the latter two processes proceed in parallel almost simultaneously, leading to a rapid decay of the K^+^ outward current, without the intervention of the hERG ‘endogenous blocker’, which is in charge of the plugging of the central pore. The hERG potassium channel fast inactivation is considered to be similar to the *Shaker* ‘C-type inactivation’ [12], which consists of a collapse of the central pore; this collapse is positioned in the proximity of the selectivity filter, whose molecular arrangement is distorted, determining its inactivation [13,14]. Under these conditions, the selectivity filter is said to be ‘inactive’ and denoted by I. The conformation of any voltage-gated tetrameric ion channel can be conveniently described by specifying between parentheses the O or C state of the activation gate, followed by the O or I state of the selectivity filter, and separating the two states with a slash. This notation, adopted in [15,16,17], is clear and straightforward. 

A repolarization induced by a sufficiently negative potential step is rapid in the hERG channel (*fast repolarization*), causing the inactive selectivity filter (sometimes called the ‘inactivation gate’ [18]) to open almost instantaneously by removal of the collapsed configuration of the central pore and of the nearby selectivity filter. However, the resulting (O/O) conformation must deal with an almost concomitant *slow deactivation*. The latter slowly moves the endogenous blocker, which comes into play for the first time since the initial slow activation and slowly fits into the central pore from the open activation gate. This blocker consists of the hERG N terminus and is similar to the ‘ball-and-chain’ motif of the *Shaker* N terminus. Of the ~390 residues of the hERG N terminus, the first 135 form the ‘eag domain’, which bears a net positive charge and whose removal results in active hERG K^+^ channels with altered gating properties [19]. This strongly suggests that, here too, like in all other known tetrameric cation channels [20,21,22], the determinant of central pore plugging by the blocker resides in the net positive charge of the latter. Deletion and point mutation studies [19,23] in the N-terminus domain also concur in identifying this domain with the endogenous blocker. 

## 2. Transitions between Contiguous Conformational States of hERG 

To describe the different conformational transitions adopted in hERG studies, it is convenient to denote the conformations with the symbolism specified in Section 1. Thus, a depolarizing pulse first triggers a *slow activation* (C/O) → (O/O), accompanied almost instantaneously by a *fast inactivation* (O/O) → (O/I). The subsequent repolarization (O/I) → (O/O) is very fast (*fast repolarization*), almost instantaneously accompanied by a *slow deactivation* (O/O) → (C/O), as shown schematically in Figure 2. 

### Quantitative Probability vs. Potential Curves

The dependence of the sole slow activation upon the depolarizing potential *ϕ*_1_, deprived of the subsequent rapid inactivation, can be determined from the K^+^ inward current peak obtained by following the slow activation at *ϕ*_1_ with a subsequent repolarization at a more negative fixed potential *ϕ*_2_ that rapidly removes the inactivation. The plot of the negative peak (tail) current at *ϕ*_2_ as a function of the progressively increasing depolarizing potential *ϕ*_1_, normalized to its maximum positive limiting value, yields a sigmoidal curve (the *steady-state activation curve*) that can be fitted with a Boltzmann function. This curve can be treated as an ‘open probability’ for the K^+^ outward current, which must be gradually damped by the concomitant fast inactivation. In fact, the K^+^ outward current attained at the end of progressively increasing depolarizing pulses *ϕ*_1_ of constant duration from a fixed holding potential, plotted against *ϕ*_1_, yields a bell-shaped current-potential curve (see the green curve in Figure 3 as an example). The decreasing branch of the bell-shaped curve is induced by a progressive increase of the fast inactivation with increasing *ϕ*_1_, which also induces a smaller decrease in the initial rising portion of the bell-shaped curve with respect to the corresponding steady-state activation curve. The increase in the fast inactivation with an increase in *ϕ*_1_ can be ascribed to a progressive increase in the collapse of the selectivity filter and of the nearby central pore induced by the increase in the resulting transmembrane potential. 

In Figure 9B (p. 1411) of [1] and in Figure 2 of [24], the steady-state activation curve is plotted together with the corresponding bell-shaped curve. The ratio of the current of the bell-shaped curve to that of the corresponding steady-state activation curve at the same transmembrane potential *ϕ* decreases almost linearly with increasing *ϕ* and can be defined as a *damping factor*. Typically, the plot of this damping factor against the transmembrane potential decreases gradually as the transmembrane potential varies from −40 to +40 mV, as shown in Figure 3, and is almost the same no matter if extracted from [1] or [24]. The equation of the straight line expressing the damping factor (blue straight line in Figure 3) is as follows: *y* = −0.00944 × *ϕ* (in mV) + 0.472.(1)

In the available literature on the hERG K^+^ channel, the bell-shaped current-potential (*I*-*ϕ*) curve is often simulated by multiplying the steady-state activation curve by an ‘activation time constant’ *τ*_a_ = *k* exp(−*Bϕ*), where *k* and *B* are positive and measured in ms and mV^−1^, respectively [25]. With both procedures, the K^+^ outward current-time (*I*-*t*) curves show an increasing inward concavity and a progressive depression the more positive *ϕ* is [4,24]. 

A repolarization induced by a sufficiently negative potential step is rapid in the hERG channel, causing the inactive selectivity filter (sometimes called ‘the inactivation gate’ [18]) to open almost instantaneously through the removal of the collapsed configuration of the central pore and of the nearby selectivity filter. However, the resulting (O/O) conformation must deal with the almost concomitant slow deactivation. The latter slowly moves the endogenous blocker, which comes into play for the first time since the initial slow activation and gradually fits into the central pore from the open activation gate. To exclude the latter effect, in principle the slow deactivation phase might be extrapolated back to the very instant when the transmembrane potential is repolarized, and the inactivation manifested during the depolarization phase is completely removed. In practice, however, this procedure is prevented by the gradual increase in the slow deactivation with a negative shift in the transmembrane potential *ϕ,* causing a decrease in the *recovery from inactivation.* The gradual loss of recovery from inactivation with a negative shift of *ϕ* can be determined using a triple pulse voltage protocol [1,8]. Typically, the cell is depolarized to *ϕ*_1_ = +40 mV for 500 ms to ensure that all channels are inactivated. They are then subjected to a series of short *ϕ*_2_ pulses (typically, +20 mV to −150 mV for 30 ms). The peak current is then measured at a *ϕ*_3_ ‘revelatory step’ (typically, +20 mV) to estimate the extent of inactivation of the ion channels at each *ϕ*_2_ potential. The plot of the percentage of inactivation against *ϕ*_2_ yields a sigmoidal curve that tends toward unity at far negative potentials (called *steady-state inactivation curve* or, more rarely, ‘relative inactivation’ [8]) and can be fitted with a Boltzmann function. This percentage decrease of recovery from inactivation can be subtracted at each applied potential from the inward K^+^ currents estimated upon ignoring it. In Liu et al. [25], the deactivation time course is fitted with a ‘deactivation time constant’ *τ*_d_ = *k’* exp(*B’ϕ*), where *k’* and *B’* are positive and expressed in ms and mV^−1^. More frequently, a biexponential deactivation time constant is conveniently adopted [24,26], with a fast and a slow component possibly originating from distinct mechanistic processes. 

The negative peaks of the experimental hERG inward K^+^ currents, which contribute to the depolarization of the cardiac action potential by injecting positive charges on the intracellular side of the membrane, increase progressively with a gradual increase in the transmembrane potential *ϕ*. The plot of these peaks against *ϕ* has a sigmoidal shape and can be fitted with a Boltzmann function; it can be regarded as the *open probability of the inward K^+^ current*. This inward K^+^ current, which gradually decreases in passing from the (O/O) to the (C/O) conformation, increases with a positive shift in the transmembrane potential *ϕ*, in apparent contrast to both the electric and osmotic forces, which would concur in moving K^+^ ions outwards. Evidently, a structural feature of the hERG channel comes into play, reversing this trend by moving K^+^ ions inwards. 

A logical explanation for this feature is offered by the appreciable narrowing of the hERG central pore, with a radius < 1 Å and hydrophobic sacks surrounding it, as revealed by a cryo-EM structure of the open hERG [3]. This sets up a barrier to the outward movement of K^+^ ions. In this case, a positive transmembrane potential *ϕ* generates a positive diffuse layer adjacent to the intracellular surface of the membrane, as well as a negative diffuse layer adjacent to its extracellular surface, to maintain the global electroneutrality of the whole interphase between the bulk intra- and extracellular media. It is the positive diffuse layer adjacent to the intracellular surface of the membrane that moves K^+^ ions inwards. Clearly, this positive potential difference is much less than that across the membrane thickness, which would favor a much more rapid outward movement of K^+^ ions if it were not prevented by the narrowing of the central pore. This is indeed a favorable feature that increases the durability over time of the slow deactivation. Since the initial outward K^+^ current is not affected by such a blockage, the conformational change determining the narrowing of the central pore is expected to take place just before the (O/O) conformation responsible for the inward K^+^ current. 

## 3. Modeling of hERG

The available literature on the kinetics and mechanism of the hERG potassium channel is based on an arbitrary interpretation of the groundbreaking results obtained by Hodgkin and Huxley [27] in their investigation of squid axon sodium and potassium currents. The heavy parametrization of their results was believed to indicate that any proposed sequence of closed states must terminate with a single open state. This ‘dogma’ of Hodgkin and Huxley parametrization has influenced the formulation of countless mechanistic models of voltage-gated tetrameric ion channels. These models are mainly stochastic and require a high number of free parameters and of unspecified conformational states [17]. Some models hypothesize that an inactivated state may stem not only from an open state at strongly depolarized potentials, but also from pre-open closed states at hyperpolarized or modestly depolarized potentials (‘closed-state inactivation’) [28]. The fact that the same set of parameters and initial conditions can yield an ensemble of different outputs imparts an inherent flexibility to stochastic models, allowing them to interpret a great number of experimental features, albeit at the expense of a high number of free parameters. Moreover, the introduction of too many states in stochastic models makes it practically impossible to ascribe a clear conformation to all of them. 

Unlike the other tetrameric ion channels, hERG exhibits outward and inward K^+^ currents, both elicited by progressively increasing depolarizing potentials *ϕ*. Consequently, the corresponding stochastic models must introduce two open states: one for the K^+^ outward currents and the other for the inward ones [29]. However, even in this case, a quantitative analysis of the activation and inactivation kinetics of the hERG channel requires a high number of free parameters (e.g., ten for the activation and eight for the inactivation in [29]). A closed-state inactivation was also hypothesized by Kiehn et al. [30] for an hERG K^+^ channel. 

We have shown that the main features of the squid axon [31] and *Shaker* potassium channels [20], the squid axon sodium channel [21], and the Ca_v_3.1 calcium channel [22] can be simulated with two or, at most, three free parameters by assuming a stepwise outward movement of the four S4 segments of these tetrameric ion channels, which determines a concomitant stepwise increase in the flow of the permeant cation. This very low number of free parameters also minimizes the number of distinct conformational states (four in the schematic of Figure 2), which can be clearly defined by simply specifying the open or closed state of the activation gate and the open or inactive state of the selectivity filter. In support of this stepwise mechanism, the conductance of the I_Ks_ ion channel pore, which provides a reserve of cardiac action potential repolarization at times of stress, was recently proposed to result from the outward movement of individual voltage sensing domains [32]. We will follow an analogous procedure with the hERG channel, using two distinct computations, one for the outward K^+^ currents and the other for the inward K^+^ currents, with the latter starting from the onset of a fast repolarization unaffected by the slow deactivation. This hypothetical situation is closely approached by inward K^+^ currents at transmembrane potentials where the steady-state inactivation curve is already very low. 

Both outward and inward K^+^ currents of the hERG channel can be simulated with procedures analogous to those adopted for the *Shaker* K^+^ channel [20]. The opening of tetrameric K^+^ ion channels, no matter if *Shaker* or hERG, involves the stepwise outward movement of its S4 segments, which pass from an ‘in’ state to an ‘out’ state. In view of the known structure of tetrameric ion channels [6], it is reasonable to expect that the stepwise outward movement of each S4 segment will be transmitted to the corresponding S6 segment of the bundle crossing constituting the activation gate, eliciting a K^+^ flow through this partially open gate. 

We will assume that this ion flow increases in proportion to the number of ‘out’ S4 segments. It is worth noting that it is only thanks to this assumption that the deterministic model succeeds in interpreting the shape and potential dependence of families of current-time (*I*-*t*) curves using only two or, at most, three free parameters. This assumption conflicts with a dogma for most electrophysiologists, according to which an ion channel opens only after all four subunits (or domains) of the ion channel are activated [17]. Conversely, in our view, an ion channel opens gradually via the progressive outward movement of the four S4 segments participating in its tetrameric structure, giving rise to an increasing ‘out’ cluster. This clustering will be referred to as an *aggregation*, to signify a progressive building up of the open channel. 

The models used to interpret all salient features of squid axon Na^+^ [21], squid axon K^+^ [31], and *Shaker* K^+^ channels [20], as well as the Ca_v_3.1 calcium channel [22], are two-state models, in that they consider the S4 segments of the four voltage sensing domains in either an ‘in’ or ‘out’ position with respect to the extracellular side of the membrane. Nonetheless, this two-state model is not rigid, because the four S4 segments of a channel are allowed to pass from the ‘in’ to the ‘out’ position step by step. From now on, to simplify the notation, the subunits of a K^+^ channel with their ‘out’ (‘in’) S4 segment will be briefly called ‘out’ (‘in’) subunits. The ‘in’ subunits are formally regarded as non-aggregated and the ‘out’ subunits as aggregated, even though they are part of the same tetrameric structure. 

The outward movement of ‘in’ subunits is simulated based on two quite reasonable assumptions. The first assumption consists in ascribing a circular shape to the cross-sectional area *A* of the ‘out’ subunits that are aggregated after moving outwards. This assumption is combined with a second reasonable assumption, according to which the rate of growth of the radius *R* of an ‘out’ cluster is proportional to the frequency of the *favorable* impacts of the cluster with the ‘in’ subunits that are moving outwards. By favorable impacts we mean those impacts in which the lateral orientation of the ‘in’ subunits that are trying to join an ‘out’ cluster by moving outwards complies with that of the subunits composing the progressively growing ‘out’ cluster. These two assumptions immediately yield a proportional relationship between the rate of growth, d*R*/d*t*, of the cluster radius *R* and the surface coverage [M] by the ‘in’ subunits. 

To this end, let us denote by *x*_1_ and *x*_2_ the mole fractions of the ‘in’ and ‘out’ subunits in the membrane, respectively; differently stated, *x*_2_ is the ratio of the number of ‘out’ subunits in the membrane unit surface area to the totality of all subunits, irrespective of their ‘in’ or ‘out’ position relative to the extracellular fluid; (*x*_1_ + *x*_2_) is clearly equal to unity. Denoting by *θ* the fraction of the membrane unit surface area covered by all subunits, irrespective of their ‘out’ or ‘in’ state, [M] is given by *θpx*_1_, where *p* is the steady-state open probability at the given depolarization potential. Upon setting *S* = *x*_2_/(*x*_1_ + *x*_2_) = *x*_2_ for convenience, we then have: [M] = *θpx*_1_ =*θp*(1 − *S*).(2)
On the other hand, the surface coverage by the ‘out’ subunits, denoted by [C], is clearly given by: [C] = *θpx*_2_ = *θpS,*(3)
and is proportional to the K^+^ outward current. 

The quantity *S* should not be confused with *p*. In fact, even though *S* attains the unit value at any given depolarization potential *ϕ* as the time *t* from the onset of the depolarizing pulse becomes sufficiently long, such a unit value refers to the maximum mole fraction of ‘out’ subunits permitted by the potential-dependent probability *p* at the given *ϕ* value, and not to the unitary mole fraction of ‘out’ subunits in the whole membrane. The latter situation is only achieved when *p* equals unity at sufficiently positive transmembrane potentials *ϕ*. It is important to reiterate that the ‘out’ or ‘in’ position of a subunit refers exclusively to its S4 segment, and not to the whole subunit, as clearly shown in Figure 4. Upon ascribing a circular shape of radius *R* to the cross-sectional area *A* of the growing ‘out’ cluster and assuming that the growth rate of *A* is proportional to the frequency of impacts of the M subunits with the circumference of *A*, it immediately follows that: (4)$$\frac{dA}{dt}=\frac{d\left(\pi R^{2}\right)}{dt}=2\pi R\frac{dR}{dt}=k_{R}2\pi R\left[M\right]\;\;\to \;\;\frac{dR}{dt}\equiv v_{R}=k_{R}\left[M\right].$$

Here, *k*_R_ is the proportionality constant for the radial growth of the ‘out’ clusters over time. 

The radial growth expression in Equation (4) can be inserted into the well-known formalism of the kinetics of nucleation and growth, which requires only one free parameter, i.e., the product of the nucleation rate constant (*k*_N_) and the square of *k*_R_ [20,21,22,31]. In practice, in all investigated systems relying on the above expression for the rate of radial growth [20,21,22,31], the best fit by the model to the experimental behavior is attained by setting the number of ‘out’ subunits composing the ‘nucleus’ equal to unity. This simply means that, in practice, no real nucleation process occurs. In this case, the nucleation rate can be interpreted as the rate at which an ‘in’ subunit M moves outwards, and can be regarded as proportional to the surface coverage [M] by the ‘in’ subunits:(5)$$dN/dt\equiv v_{N}=k_{N}\left[M\right],$$
where *k*_N_ is a proportionality constant. 

The kinetic treatment of aggregation starts from considering that d*N* in Equation (5) expresses the number of ‘in’ subunits per unit surface area of the membrane that are moving outwards in an infinitesimal time interval between *y* and *y +* d*y*, interposed between the onset (*t* = 0) of the depolarizing pulse and a given ‘observation time’ *t*. Upon again ascribing a circular shape to the clusters, the area *A* of the membrane unit surface covered by the clusters generated by d*N* during the finite time interval between *y* and *t* is given by: (6)$$dN{\int_{0}^{A(t)}dA=\pi \left(\int_{0}^{R}dR\right)}^{2}dN=\pi {\left[\int_{y}^{t}{\left(dR/dt\right)}_{z}dz\right]}^{2}dN.$$

The parameter *z* is an auxiliary variable having the dimensions of time, which is allowed to vary between *y* and the observation time *t*. Upon integrating Equation (6) over time between the limits of integration *y* = 0 and *y* = *t*, we obtain the ratio of the membrane area covered by the ‘out’ clusters (eventually yielding the ion channels) to that covered by all subunits, no matter if aggregated or not: (7)$$S_{x}=\pi {\int_{0}^{t}dy\left[\int_{y}^{t}{\left(dR/dt\right)}_{z}dz\right]}^{2}{\left(dN/dt\right)}_{y}\equiv \pi {\int_{0}^{t}dy\left[\int_{y}^{t}v_{R}\left(z\right)dz\right]}^{2}v_{N}\left(y\right).$$

The above dimensionless quantity is equal to the ratio, *S*, defined in Equation (2). However, the integration of Equation (6), leading to Equation (7), ignores the possible overlapping of the progressively growing clusters. Hence, the resulting quantity *S*_x_, called *extended area*, is greater than *S* and is distinguished from it by the subscript *x*. To obtain *S* from *S*_x_, a simplified approach leading to the rigorous expression derived by Avrami [33,34] is as follows. We will make the reasonable assumption that the growth rate of *S*_x_, d*S*_x_/d*t*, is proportional to the total unit surface area of the membrane, *S*_T_ = 1, whereas the growth rate of *S*, d*S*/d*t*, is proportional to the fraction (*S*_T–_*S*) of the unit surface area of the membrane still unoccupied, according to the same proportionality constant, *k*_g_. Consequently, we have: (8)$$\frac{dS_{x}}{dt}=k_{g}S_{T};\;\frac{dS}{dt}=k_{g}\left(S_{T}-S\right)\;\to \;\frac{dS}{dt}=\left(1-S\right)\frac{dS_{x}}{dt}\;\mathrm{with}:\;S_{T}=1.$$
The last differential equation in Equation (8) allows *S* to be calculated from *S*_x_. 

Replacement into Equation (7) of *v*_R_ and *v*_N_ from Equations (4) and (5), with [M] = *θp*(1 − *S*) in view of Equation (2), and repeated differentiation of the resulting equation with respect to time *t* via the generalized Leibnitz formula yields the following differential equations: (9)$$\frac{dS_{x}}{dt}=2\pi k_{R}pf_{1}f_{2};\;\frac{df_{2}}{dt}=k_{R}pf_{1}f_{3};\;\frac{df_{3}}{dt}=k_{N}pf_{1}\;\mathrm{with}:f_{1}\equiv \theta \left(1-S\right).$$

The system of the three differential equations in Equation (9) and of the additional differential equation in Equation (8) can be solved by the fourth-order Runge–Kutta method, allowing the calculation of *S* as a function of time *t* and of the transmembrane potential *ϕ* [20,21,22]. 

It is only at this point that hERG starts differing from *Shaker* by exhibiting a fast inactivation (O/O) → (O/I), which is concomitant with the slow activation and is expressed by the experimental damping factor of Equation (1). Hence, the outward K^+^ current of hERG is obtained by multiplying the outward current proportional to [C] in Equation (3) by the damping factor in Equation (1), yielding:
*outcurr* ∝ *θpS* × (−0.00944 × *ϕ* + 0.472). (10)

Here, *p* is the open probability (i.e., the steady-state activation curve normalized to unity). 

The parameter *θ* is not a free parameter. In fact, it can be ignored by formally setting it equal to unity and using a suitable conversion factor to compare a family of calculated K^+^ outward current vs. time (*I*-*t*) curves at different transmembrane potentials with the corresponding experimental ones. The well-known formalism of the kinetics of ‘nucleation and growth’ [20,22,31,35] requires only a single free parameter, i.e., the product of the nucleation rate constant *k*_N_ and the square of the rate constant *k*_R_ of growth of the radius of the cross-sectional area of the clusters of ‘out’ subunits. In what follows, *k*_N_ and *k*_R_ will be constantly set equal to 10^2^ s^−1^ and 10^5^ s^−1^, i.e., two orders of magnitude smaller than those in the *Shaker* potassium channel [20]. The set of the four differential equations in Equations (8) and (9) can be solved by the Runge–Kutta method of the fourth order. 

At this point, a distinction must be made between the K^+^ outward current, which is not affected by the endogenous blocker, and the K^+^ inward current. The mathematical treatment underlying the derivation of the K^+^ inward current is identical with that used for the outward K^+^ current, except for the change in the sign of the current and the notable effect induced by the intervention of the endogenous blocker. 

We already said that the endogenous blockers of *Shaker* K^+^, Na^+^, and Ca_v_3.1 cation channels bear a net positive charge. This is also the case with the hERG potassium channel N terminus [19]. Upon observing that all endogenous blockers bear a net positive charge, during the slow deactivation phase they are expected to be progressively attracted along the central pore by the increasingly negative charge that the step-by-step outward movement of the four S4 segments leaves behind on the intracellular side of the hydrophobic gating pores in the corresponding voltage sensing domains. The average number density, *N*_bk_, of blockers bound to the receptor, which is located within the pore and close to the selectivity filter, is related to the total number density of blockers, *N*_bk,t_, in the membrane via a one-sided Boltzmann-like function [21]: (11)$$N_{\mathrm{bk}}=N_{\mathrm{bk},t}/\left[1+\exp\left(W_{\mathrm{bk}}/kT\right)\right].$$

Here, *W*_bk_ is the work of formation of bound blockers. It consists of a structural component, *U*_bk_, and of a Coulomb-attractive component proportional to the time dependent charge, −*q*, progressively left on the cytoplasmic side of the gating pores by the stepwise outward movement of the four S4 segments, multiplied by the constant positive charge of the blocker: *W*_bk_ = *U*_bk_ − *k*″*q*.(12)

The parameter *k*″ is a positive proportionality constant for the Coulomb-attractive interaction. Since the blocker is immersed in the intracellular fluid before folding into the channel structure, the effect of the transmembrane potential is negligible and can be ignored. If we make the reasonable assumption that the structural component *U*_bk_ is much greater than *kT*, Equation (11) simplifies as follows: *N*_bk_ ∝ exp(*k*″*q*/*kT*) ∝ exp(*k*′*S*/*kT*).(13)

In the above equation, account is taken of the fact that the charge −*q* progressively left on the cytoplasmic side of the gating pores by the stepwise outward movement of the four S4 segments is equal in magnitude and opposite in sign to the positive charge moved by the S4 segments passing across their relative gating pores, i.e., the true positive gating charge *q*. Conversely, *S* is the total gating charge normalized to unity defined in Equation (2), which attains the unit value for sufficiently long times at all depolarization potentials, and hence, it is proportional to *q* according to a proportionality constant. The product of the latter proportionality constant and *k*″ yields a different proportionality constant *k*′. 

The ‘driving force’ of the slow deactivation can be regarded as proportional to the difference between the maximum value attained by *N*_bk_ at sufficiently long times, when *S* equals unity, and its value at any given time *t*. From Equation (13) and upon normalization to unity, we have: (14)$$\frac{\exp\left(\frac{k^{\prime }}{k_{B}T}\right)-\exp\left(\frac{k^{\prime }S}{k_{B}T}\right)}{\exp\left(\frac{k^{\prime }}{k_{B}T}\right)}=1-\exp\left[-k_{h}\left(1-S\right)\right]\;\;\mathrm{with}:\;k_{h}\equiv \frac{k^{\prime }}{k_{B}T}.$$

Since the ion flow takes place on the fraction of the membrane unit surface area covered by the open ion channels, [C] = *θpS* (see Equation (3)), the K^+^ inward current, *incurr*, is proportional to this fraction multiplied by the driving force of Equation (14): (15)$$\mathrm{incurr}\;\propto \;\theta pS\;\left\{1-\exp\left[-k_{h}\left(1-S\right)\right]\right\}.\;$$

As *S* increases gradually over time from 0 to unity following a sufficiently long depolarizing pulse, the first factor of proportionality, *θpS*, increases rapidly over time and is responsible for the initial rising section of the K^+^ inward ionic current. Conversely, the second factor of proportionality, $\left\{1-\exp\left[-k_{h}\left(1-S\right)\right]\right\}$, decreases much more slowly and accounts for the slow exponential decay of the K^+^ inward current, typical of hERG slow deactivation. The current decay is ascribed to the stepwise outward movement of the endogenous blocker along the central pore, which is induced by the concomitant stepwise outward movement of the S4 segments of the four hERG subunits, gradually plugging the central pore. 

Here, *k*_h_ is a free dimensionless parameter required to simulate the exponential decay of the K^+^ inward current. The probability *p* in Equation (15) is the experimental ‘open probability of the inward K^+^ current’, which contributes to the depolarization of the cardiac action potential by injecting positive charges on the intracellular side of the membrane. The simulation of the K^+^ inward current requires only two free parameters, i.e., the overall rate constant of aggregation of the four ‘out’ subunits, *k*_N_*k*_R_^2^, and the free parameter k_h_ expressing the rate constant for the exponential decay of the K*^+^* inward current. 

### Simulation of the hERG Potassium Channel in Piper et al. [8]

We will now examine in detail the paper by Piper et al. [8], which provides the experimental values of both outward and inward K^+^ currents while maintaining physiological K^+^ concentrations in both the intracellular and extracellular solutions. The five transmembrane potentials for both the outward and inward K^+^ currents (i.e., −50, −30, −10, +10, and +30 mV) are unequivocally ascribable to these currents by using five different colors for the five *I*-*t* curves. This paper provides the Boltzmann parameters of the experimental steady-state activation curve in Figure 2c of [8] (*z* = 2.4, *ϕ*_1/2_ = −5.6 mV, *a* = exp[−*zϕ*_1/2_/25.6 mV] = 1.69), and those of the experimental steady-state inactivation curve in Figure 4a of [8] (*z* = 1.1, *ϕ*_1/2_ = −82 mV_,_
*a* = 33.9). Piper et al. [8] also provide the Boltzmann parameters for the ‘open probability of the inward K^+^ current’ predicted by their stochastic model in Figure 5f of [8] (*z* = 2.2, *ϕ*_1/2_ = −9 mV_,_
*a* = 2.16).

In view of the substantial independence of the inward currents from the outward ones, sanctioned by the very fast repolarization (see also Figure 2), they can be treated by compiling two separate Fortran text files. The K^+^ outward *I*-*t* curves are calculated from Equation (10). Upon using the experimental steady-state activation curve as the open probability, the resulting calculated K^+^ outward *I*-*t* curves in Figure 5b only require the single free parameter expressing the overall rate constant *k*_N_*k*_R_^2^ = 1 × 10^12^ s^−3^ of cluster aggregation. They are moderately close to those calculated by Piper et al. in Figure 5b (right) of [8] and shown in Figure 5a below, whose computation with the stochastic model requires more than 40 free parameters, as summarized in Table 1 of [8]. 

The K^+^ inward *I*-*t* curves in Figure 6b are calculated from Equation (15) with the same rate constant *k*_N_*k*_R_^2^ = 1 × 10^12^ s^−3^ of cluster aggregation used in Figure 5 and with *k*_h_ = 1. They are quite close to those calculated by Piper et al. in Figure 5b (right) of [8]. The open probability *p* in this equation is the ‘open probability of the inward K^+^ current’ estimated in [8]. 

At the holding potential of −110 mV, the hERG ion channel is closed and starts to open at about −50 mV, where the steady-state inactivation curve in Figure 4a of [8] is already relatively low. This is even more true for the K^+^ inward *I*-*t* curves in Figure 6b at more positive transmembrane potentials, where the steady-state inactivation curve becomes vanishingly small. The K^+^ ions are pushed inwards rapidly, but the open channel tends to close more slowly due to the endogenous blocker, which sneaks into the activation gate, inducing slow deactivation.

A peculiar feature is reported in Figure 1b of [8], where the inward K^+^ current obtained by stepping from a holding potential of −110 mV decreases as the transmembrane potential *ϕ* becomes progressively less negative and becomes outward as soon as *ϕ* becomes positive. Evidently, this behavior, which is consistent with the combined effect of electric and osmotic forces, strongly suggests a disruption of the structural feature described by Wang and MacKinnon in [3] and responsible for the K^+^ inward *I*-*t* curves in Figure 6. 

The present deterministic model allows gating charges to be easily estimated starting from the parameter *S* defined in Equation (2), which measures the mole fraction of ‘out’ S4 segments and tends toward unity with an increase in the length of the depolarizing pulse. This is just a measure of the gating charge normalized to unity, and its time derivative yields the corresponding gating current. As an example, Figure 7 shows the gating current elicited by a 300 ms long square wave pulse from −110 to −10 mV and vice versa. The shoulder following the initial positive peak might possibly be induced by the slow activation. 

The hERG potassium channel contributes to depolarizing the neuron membrane with its K^+^ inward current and to repolarizing it with its K^+^ outward current. The initial flat plateau of the cardiac action potential is determined by a small K^+^ outward current that rapidly undergoes fast inactivation. A contribution to the initial depolarization comes from sodium and calcium ions, which move from the extracellular fluid to the cytosol [36]. The subsequent rapid repolarization of the terminal portion of the cardiac action potential proceeds quickly due to the inactivation of calcium and sodium ion channels, which determines a decrease in the intracellular movement of positive charges; this effect is combined with a rapid repolarizing outward K^+^ current of hERG and with a slow delayed rectifier *I*_Ks_ potassium current [37]. These currents cause an increase in the movement of positive charges out of the cell, which determines a current peak of the action potential and has a role in determining action potential duration. The increasingly more outward overall membrane current causes the cardiac action potential to move to its resting value, with a decrease in the electrochemical driving force for K^+^ efflux and the slow deactivation of hERG [38]. Loss-of-function mutations of the hERG gene prolong this final repolarization, inducing an anomalously long time spent by the heart muscle to contract and relax, i.e., the so-called ‘QT prolongation’. 

## 4. Conclusions

The available literature on the modeling of hERG adopts Markov stochastic models, which assume that any proposed sequence of closed states of a voltage-gated tetrameric ion channel terminates with a single open state; this ‘biophysical dogma’ [17] has influenced all the pertinent literature and imposes the use of more than ten free parameters to provide quantitative models of the experimental behavior, making it impossible to ascribe clearcut conformations to the numerous states involved. 

The present work provides the first deterministic model of hERG functional activity. This model assumes that a stepwise outward movement of the four S4 segments of hERG is accompanied by a concomitant stepwise increase in the K^+^ ion current, allowing the K^+^ outward and inward current-time (*I*-*t*) curves to be predicted with one (or, at most, two) free parameter. This result is not only significant in and of itself. It also allows the total number of conformations involved in both depolarization and repolarization to be reduced to four, as shown in Figure 2, providing a clearcut picture of the whole hERG functional activity. *Mutatis mutandis*, this approach is similar to the one that was already used to account for all salient features of the *Shaker* K^+^ channel [20], the squid axon Na^+^ channel [21], and the Ca_v_3.1 calcium channel [22]. 

This work also points out for the first time the fundamental role played by a peculiar structural feature revealed by a recent cryo-EM structure of an open hERG [3], where the central pore exhibits an atypically small cross section of radius < 1 Å midway along its length, surrounded by four deep hydrophobic pockets. It is this narrowing that prevents the K^+^ current during slow deactivation from moving outwards under the combined effect of electric and osmotic forces, by blocking and diverting it downwards using the positive electric field generated by the positive diffuse layer located on the intracellular surface of the cell membrane. 

## Figures and Tables

**Figure 1 molecules-28-03514-f001:**
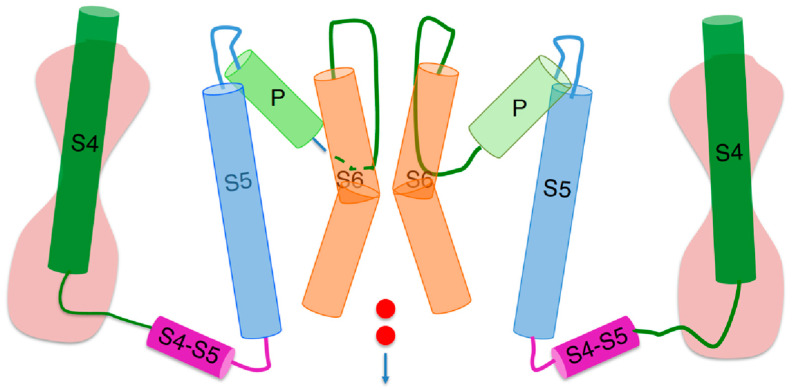
Schematic picture of two opposite subunits of a peculiar hERG open conformation [3], simplified by removing the S1–S3 helices of each voltage sensing domain and replacing them with an hourglass symbolizing two hydrophilic vestibules separated by the gating pore. The cylinders denote α-helices, whereas the curved lines connecting the cylinders denote intra- or extracellular loops. The two small red circles symbolize an inward potassium current. Extracellular fluid at the top, intracellular medium at the bottom.

**Figure 2 molecules-28-03514-f002:**
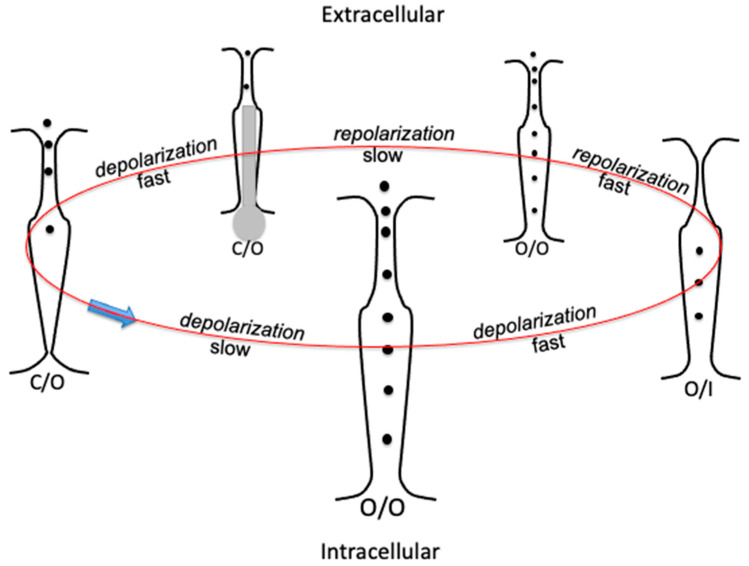
Schematic picture of the transverse section of the hERG K^+^ channel, illustrating a gating cycle of its consecutive conformational states by arranging them in a circle: selectivity filter at the top, activation gate (the bundle crossing) at the bottom. States C/O, O/O, and O/I are indicated at the bottom of each transverse section. Transition rates are indicated by the words “fast” and “slow” along the gating cycle. The endogenous blocker is denoted in grey.

**Figure 3 molecules-28-03514-f003:**
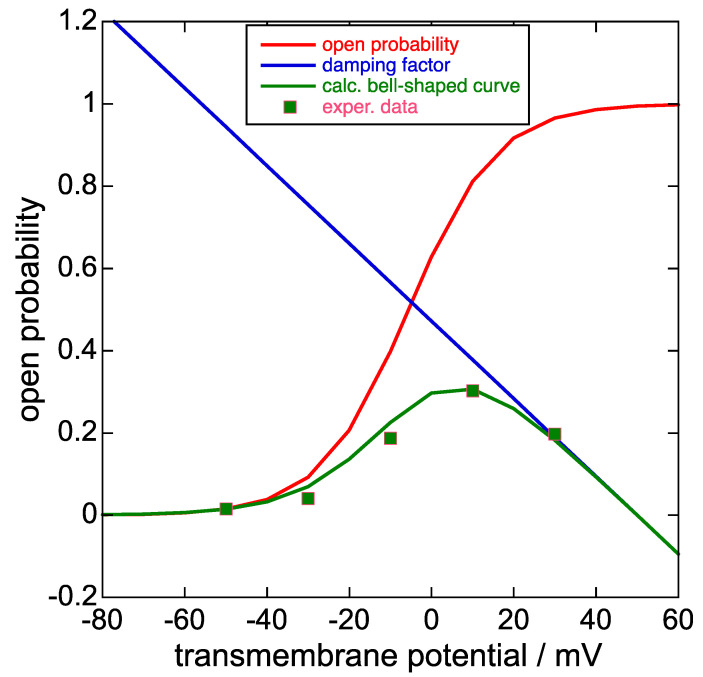
Plot of the steady-state activation curve, i.e., the open probability (red curve), the damping factor (blue straight line), the product of the two at each transmembrane potential (green curve), and the experimental points of the bell-shaped curve in Figure 1c of [8] upon ascribing a 0.30 value to its highest point on the vertical scale normalized to unity. Unfortunately, Piper et al. [8] did not show this bell-shaped curve together with the corresponding steady-state activation curve on the same vertical scale, unlike [1] and [24]. The damping factor does not strictly refer to the open probability scale, although it takes values between about 0 and 0.8 on this scale for transmembrane potential values *ϕ* varying from +40 to −40 mV.

**Figure 4 molecules-28-03514-f004:**
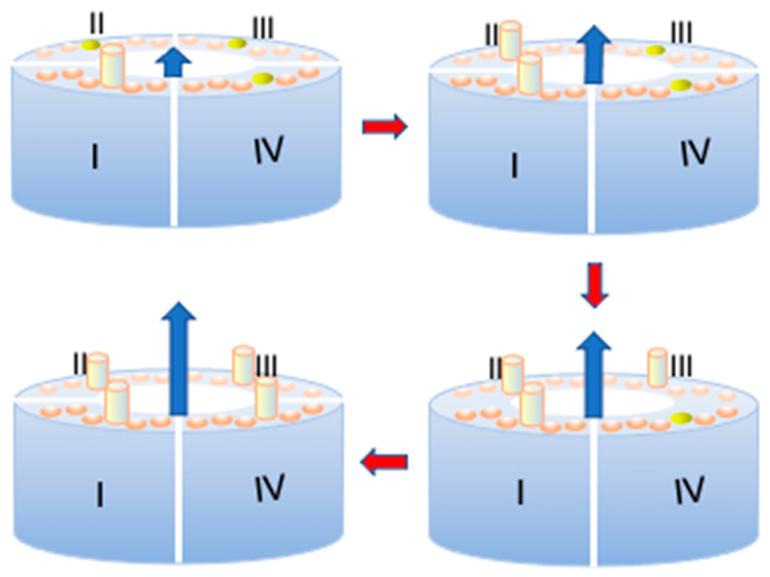
Schematic of the arrangement of the four subunits (I)–(IV) of a *Shaker* or hERG potassium channel around the central pore during the stepwise opening of the channel. The six disks on top of each of the four subunits denote the six α-helices, with the S4 helices in yellow and the other five helices in light brown. More precisely, during the first opening step of the channel, the S4 helix of subunit I moves outward and is represented as a yellow cylinder, while the three other S4 helices are still in the ‘in’ position. Following the direction of the red arrows, all four S4 α-helices move outwards step by step, and the K^+^ outward current, represented as a progressively longer outward blue arrow, increases accordingly.

**Figure 5 molecules-28-03514-f005:**
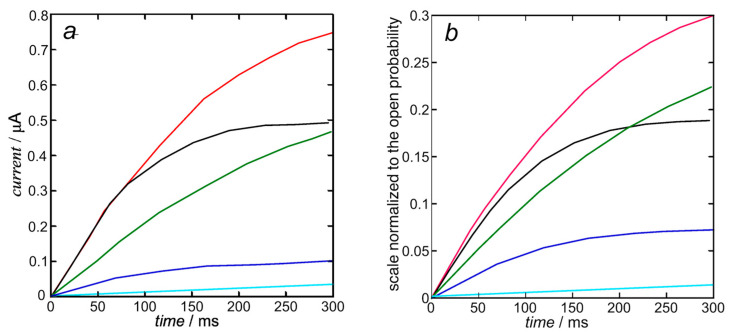
Experimental K^+^ outward *I*-*t* curves taken from Figure 5b (left) of [8] (**a**) and those calculated from Equation (10) (**b**) with *k*_N_*k*_R_^2^ = 1 × 10^12^ s^−3^ at +30 mV (black), +10 mV (red), −10 mV (green), −30 mV (blue), and −50 mV (cyan). The calculated curves are moderately close to those calculated by Piper et al. in Figure 5b (right) of [8], since they are both based on the same experimental steady-state activation curve.

**Figure 6 molecules-28-03514-f006:**
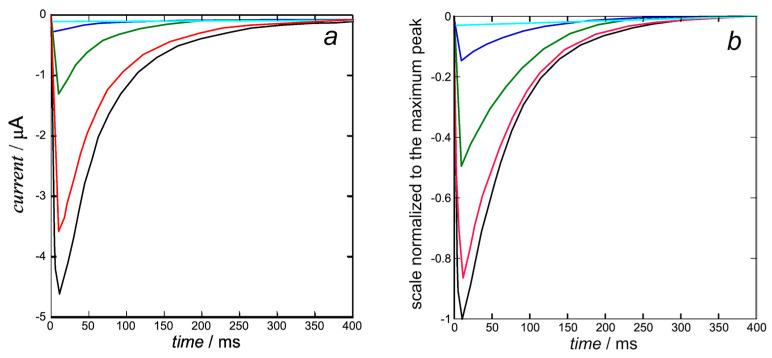
Experimental K^+^ inward *I*-*t* curves taken from Figure 5b (left) of [8] (**a**) and calculated from Equation (15) (**b**) with *k*_N_*k*_R_^2^ = 1 × 10^12^ s^−3^ and *k*_h_ = 1 at +30 mV (black), +10 mV (red), −10 mV (green), −30 mV (blue), and −50 mV (cyan). The calculated curves (**b**) are quite close to those calculated by Piper et al. in Figure 5b (right) of [8], since they are both based on the same ‘open probability of the inward K^+^ current’.

**Figure 7 molecules-28-03514-f007:**
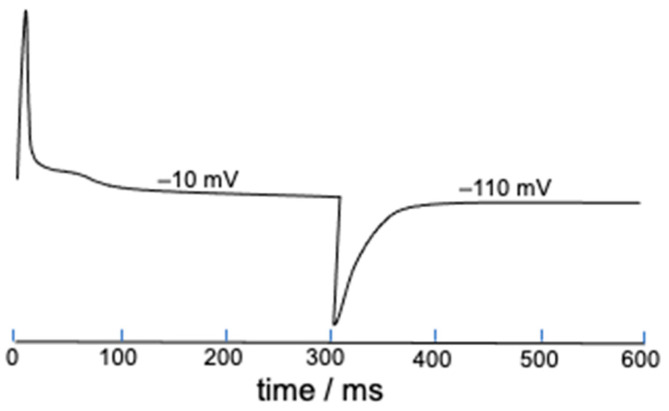
Gating current estimated from the time derivative of *S* and elicited by a 300 ms long square wave pulse from −110 to −10 mV and vice versa. The potential values on the gating current denote the final potential of the square wave pulse.

## Data Availability

Not applicable.

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
