# Peer review of "An Insight into the Potassium Currents of hERG and Their Simulation"

_molecules, 2023, doi:10.3390/molecules28083514_

Round 1
Reviewer 1 Report
The manuscript describes a novel mathematical deterministic model of hERG channel gating, which uses less variables than more traditional stochastic or Markov state models of channel gating. The author describes application of the model to both outward and inward K+ conduction obtaining reasonable agreement with one set of previously reported experimental data.
The authors do a nice job describing some of the key structural features of hERG channel, but I get a bit lost trying to follow their modeling sections. There is a lot of comparison between hERG and Shaker and sometimes it is not clear to which channel the discussion applies in particular.
The model schematics with states, transitions between them and key variables would be extremely helpful. It is also confusing whether the model applies to measuring ionic or gating current as ref. 12 Piper at al focuses on gating currents, but reports outward and inward ionic currents as well. Actually physiological relevance of inward K+ current as described by the author is not clear at all. I think there is some confusion in the manuscript on the concept of inward rectification observed in experiments with very hyperpolarized holding potentials, lower than E(K+) and their physiological role in terms of hERG K+ conduction (Ikr) during cardiac action potential. This needs substantial clarification.
I'm fairly skeptical of this model in general. The author seems to be criticizing the "biophysical dogma" that dominates the field, but I don't entirely see their point. I think I'd like to see more comparisons with experimental data. I looked at the Piper et al. study that uses a Markov formalism as well as experimental current recordings. I guess the manuscript's Figures 3 and 4 agree qualitatively with Figure 5b from Piper et al., but it's hard for me to say if this modeling fits well. Overall, I'd say this paper has an admirable goal of trying to reduce the number of free parameters in a model and better model phenomena like intermediate conductance states. I'd like to see more results than just one I-t curve for a particular voltage protocol to be convinced that this deterministic model works well.
There have been other numerous model reduction studies, for instance, Docken et al PLoS Comput Biol2021 Jun 29;17(6):e1009145 doi: 10.1371/journal.pcbi.1009145. And stochasticity is well known attribute of ion channel gating and other phenomena captured e.g. in long-scale molecular dynamics simulations (Jensen et al Science. 2012 Apr 13;336(6078):229-33. doi: 10.1126/science.1216533.), cryo-EM experiments showing multiple closed and single open conformation (Hite and MacKinnon Cell 2017 Jan 26;168(3):390-399.e11. doi: 10.1016/j.cell.2016.12.030. ) and single-channel recordings (Hille Ion channels in excitable membranes book and many other sources) There is also a good discussion about stochastic channel models and their pros and cons here: http://www.scholarpedia.org/article/Stochastic_models_of_ion_channel_gating
Reviewer 2 Report
The manuscript proposes a new model of hERG channel whereby the gates open in step-wise increments rather than an all-or-none fashion. It is claimed that the all-or-none formulation of the open state has been unduly enshrined by biological dogma and that, by doing away with that dogma, the new model can recreate the observed data with fewer parameters than existing models (2 parameters versus 10 parameters).
I am not an expert on the conformational states on the hERG channel and I found this paper very difficult to assess. It assumes a great deal of familiarity with the field, in particular several papers that are referred to without much explanation. The equations are themselves stated without much explanation. So after multiple reads, I still do not understand what exactly is being modeled. The manuscript appears to be written for a very small audience that is familiar with two or three key papers. Stylistically, it was hindered by the lack of conventional Results and Methods sections.
The basic premise (progressive gating) seems worthy of publication but in my opinion the paper needs major revisions to make it accessible to a wider audience. In particular, the equations need to be better motivated and explained. I also suggest including additional figures that help illustrate the points made in the text. An explicit counter-example of an existing model would be helpful for comparing the competing approaches too.
Minor Comments
Initially I was perplexed about the relevance of the small diameter of the central pore. As described on line 64, it only seems to have a passing relevance to drugs, which is another story. It is better explained later at line 206. Perhaps the details in lines 60-63 could be omitted/postponed to avoid confusion.
Lines 217-235: I found these paragraphs very difficult to follow. I suggest taking some time here to give more background to the existing approach so that your objections to them are clearer. For instance, it would be helpful to give the reader an exemplar equation which directly shows the free parameters in the current models that you seek to eliminate. A few illustrative figures would help too.
Lines 236: We will show …
Lines 250: The definition of S comes from nowhere and left me perplexed. What is the point of this equation? What are x1 and x2? Regrettably, I could not make any sense of this equation nor those that followed.
Line 330: … they can be treated as separate simulations.
Lines 367-371: What does it really matter if the equations are stochastic versus deterministic? It is the number of free parameters that matters most.
Lines 385-395: The final paragraph is not really a conclusion. It does not address the benefits of conceiving the hERG channel as having progressive open-states rather than the conventional all-or-none state. It merely summarizes the ionic mechanisms of the cardiac action potential.
Reviewer 3 Report
In this study a model for simulating the current across the hERG channel is described. HERG is responsible for the repolarizing IKr current in cardiac action potentials. Alterations of IKr, due to hERG mutations or drugs, might be responsible for arrythmias, eventually leading to cardiac arrest and sudden death. This important role of IKr in the cardiac activity justifies a wide interest for the hERG channel, which is likely the potassium channel that attracted more efforts from the scientific community both regarding experimental analyses and modelling. In this context, the comparison with previous literature is crucial. In particular, these following points should be proved and discussed:
1) Is the model proposed here better than previous models in reproducing experimental data? Comparison with experimental data and simulations with previous models should be included in the Figures 3 and 4.
2) Is it possible to use the proposed model for simulating action potentials with the same accuracy obtained with previous IKr models ?
3) Does the model offer novel insights into the functioning of the hERG channel ?
While point (3) is partially discussed (but could be further expanded), points (1) and (2) are completely not considered, severely reducing the potential impact of the study.
Round 2
Author Response
Please see attachment with author's detailed response

Reviewer 2 Report
Having no expertise in the conformational states of the hERG channel, I cannot comment on the technical merits of this paper. However I can say that the writing is vastly improved, particularly with regard to expanding the text (and figures) to suit a wider audience. The layout of the paper remains a little idiosyncratic in that it lacks traditional Results and Methods sections. Nonetheless, the essential information is present. So in that regard, I would say the paper is suitable for publication.
Author Response
Thanks so much for helping check the revised version.